



# Folded Tubular Photometer for atmospheric measurements of NO₂ and NO

John W. Birks[1], Peter C. Andersen[1], Craig J. Williford[1], Andrew A. Turnipseed[1], Stanley E. Strunk[1], Christine A. Ennis[1], and Erick Mattson[2]

[1]2B Technologies, Inc., 2100 Central Ave., Suite 105, Boulder, CO 80301
[2]Colorado Dept. of Public Health and Environment, Air Pollution Control Division/Technical
  Services Program, 4300 Cherry Creek Drive South, Denver, CO 80246

*Correspondence to:* John Birks (johnb@twobtech.com)

**Abstract.** We describe and characterize a modular Folded Tubular Photometer for making direct measurements of the concentrations of air pollutants such as nitrogen dioxide (NO₂), sulfur dioxide (SO₂), ozone (O₃), and black carbon particulate matter. Direct absorbance measurements using this photometer can be made across the spectral range from the ultraviolet (UV) to the near-infrared. The absorbance cell makes use of modular components (tubular detection cells and mirror cubes) that allow construction of path lengths of up to 2 meters or more while maintaining low cell volumes. The long path lengths and low cell volumes enable sensitive detection of ambient air pollutants down to low part-per-billion levels for gas species and aerosol extinctions down to 1 Mm⁻¹, corresponding to ~0.1 μg m⁻³ for black carbon particulates. Pressure equalization throughout the stages of the absorbance measurement is shown to be critical to accurate measurements of analyte concentrations. The present paper describes the application of this photometer to direct measurements of nitrogen dioxide (NO₂) and the incorporation of design features that also enable measurement of nitric oxide (NO) in the same instrument. Excellent agreement for ambient measurements along an urban roadside was found for both NO₂ and NO measured by the Folded Tubular Photometer compared to existing standard techniques. Compared to commonly used methods for measurements of NOₓ species, the advantages of this approach include 1) an absolute quantification for NO₂ based on the Beer-Lambert Law, thereby greatly reducing the frequency at which calibrations are required; 2) the direct measurement of NO₂ concentration without prior conversion to NO as is required for the commonly used chemiluminescence method; 3) the use of modular components that allow construction of absorbance detection cells of varying lengths for extending the dynamic range of concentrations that can be measured; 4) a more economical instrument than other currently available direct measurement techniques for NO₂; and 5) the potential for simultaneous detection of additional species such as SO₂, O₃, and black carbon in the same instrument. In contrast to other commercially available direct NO₂ measurements, such as cavity-attenuated phase shift spectroscopy (CAPS), the Folded Tubular Photometer provides a means for measuring NO simultaneously in the same apparatus by quantitatively converting NO to NO₂ with ozone, which is then detected by direct absorbance.





## 1. Introduction

Poor air quality related to anthropogenic activity has been estimated to contribute up to nearly 7 million premature deaths globally on an annual basis (World Health Organization, 2014). Air pollutants such as ozone ($O_3$), nitrogen dioxide ($NO_2$), sulfur dioxide ($SO_2$), and particulate matter

(PM) have been designated "Criteria Pollutants" in the United States (U.S.) because of their well-documented adverse health effects as well as their ability to damage crops and natural ecosystems. Species such as $O_3$ and PM have significant impacts on Earth's radiation balance, thus also impacting climate. Concentrations of these pollutants are routinely measured in both ambient air and from direct industrial emissions (e.g., smokestack and fenceline monitoring). Limits on both pollutant emission

and ambient concentrations are regulated by the U.S. Environmental Protection Agency (EPA). Concentrations of these pollutants are regulated by many other countries as well. There are also non-regulated species that are known to increase health risks and affect climate, such as black carbon particulates (BC, a subset of PM). Pollutant species can either be produced directly by various combustion processes (i.e., $NO_2$, $SO_2$, and BC) or formed by secondary photochemistry from other

precursor chemicals. For example, it has been known since the 1950s that ozone is a secondary pollutant formed in the interaction of sunlight with volatile organic compounds (VOCs) and oxides of nitrogen ($NO_x = NO + NO_2$) (Haagen-Smit and Fox, 1954; Birks, 1998). Nitric oxide (NO) is also of great significance since it is the principal precursor to $NO_2$ and serves as a catalyst in the atmosphere for formation of ozone. Therefore, it is expected that measurements of $O_3$, $NO_2$, NO, $SO_2$, and black

carbon will be required far into the future.

It is critical to obtain reliable, long-term interference-free measurements of these atmospheric pollutants, ideally with instruments that require little maintenance and minimal need for re-calibration. Currently, a variety of methods are used to monitor these different species, each having its own advantages and problems. Many of the current methods need frequent calibration or rely on

indirect methods. For example, $NO_2$ is most commonly measured indirectly by conversion of $NO_2$ to NO, which is then measured by chemiluminescence (Parrish and Fehsenfeld, 2000). Absorption photometry is a direct measurement technique that is based on the intrinsic absorption characteristics (wavelength-dependent absorption cross sections) of the species of interest. Ozone, $NO_2$, $SO_2$, and black carbon all absorb at various wavelengths in the ultraviolet, visible, and/or near infrared. Light

absorbance is governed by the Beer-Lambert Law:

$$\frac{[I]}{[I_o]} = e^{-\sigma l c} \quad or \quad c = \frac{1}{\sigma l} ln\left(\frac{I_o}{I}\right) \tag{1}$$

where $I_o$ is the light intensity passing through the absorbance cell with no analyte (e.g., $O_3$, $NO_2$, $SO_2$, black carbon) present, $I$ is the intensity of light passing through the absorbance cell when the analyte is present, $\sigma$ is the extinction coefficient of the analyte (absorption cross section in $cm^2$ $molec^{-1}$ for

gases; mass extinction coefficient in $m^2$ $g^{-1}$ for particulates), $l$ is the path length through the detection cell (cm or m), and $c$ is the concentration of analyte within the detection cell ($molec$ $cm^{-3}$ for gases; μg





m$^{-3}$ for particulates). Gas-phase concentrations are typically converted to mixing ratios by measuring the temperature and pressure within the absorbance cell and applying the ideal gas law. Light absorbance is an especially attractive technique, since it relies only on knowing $\sigma$, which is an intrinsic property of the molecule in the case of gas-phase species; the path length, which is easily

measured; and the ability to measure relative light intensities. Key to using absorption photometry is understanding the limits to the analytical precision (relying on the magnitude of $\sigma$, the minimum detectable absorbance, and the path length) and insuring adequate selectivity over potential interferences (by selection of analytical wavelength(s) not significantly absorbed by other species and/or by selective scrubbing of the analyte).

For ozone, the most common measurement method is by absorbance of the 253.7 nm line of a low-pressure mercury lamp. Here, co-absorbing interferences are small due to the large $O_3$ absorption cross section (e.g., Turnipseed et al., 2017). Atmospheric measurements are easily made because the required precision (low ppb) can be achieved with practical path lengths ($l$, Eq. (1)). The absorbance, $\ln(I_o/I)$, can be measured in modern photometers with a precision (standard deviation or RMS noise)

of typically ~$3 \times 10^{-6}$. Combining this with the absorption cross section and optical path length, Eq. (1) can be rearranged to determine the overall precision expected for measurement of a given analyte:

$$Precision\ (ppb) \cong \frac{3 \times 10^{-6}}{\sigma l (P/kT)} \times 10^9 \qquad (2)$$

Here, $P$ and $T$ are the absorbance cell pressure and temperature, respectively, and $k$ is the Boltzmann constant. For ozone, $\sigma_{253.7nm} = 1.15 \times 10^{-17}$ cm$^2$ molec$^{-1}$ (Burkholder et al., 2015) and the precision is

calculated to be 0.7 ppb for a path length of 15 cm or 0.35 ppb for a path length of 30 cm. These are in good agreement with the performance of commercially available ozone monitors.

Pollutants such as $NO_2$, $SO_2$, and black carbon absorb much less strongly than ozone in the spectral region where stable light sources exist ($\lambda > 250$ nm), thus requiring much longer path lengths. For $NO_2$, with an absorption cross section of ~$6 \times 10^{-19}$ cm$^2$molec$^{-1}$ at 405 nm (near the peak of the

$NO_2$ absorption spectrum; Burrows et al., 1998) and assuming the same minimum measurable absorbance ($3 \times 10^{-6}$), a path length of ~ 203 cm is required to obtain a precision of 1 ppb. This is similar to that of $SO_2$ if measured at 290 nm ($\sigma \sim 7 \times 10^{-19}$ cm$^2$ molec$^{-1}$, Vandaele et al., 1994). The mass extinction coefficient used for black carbon absorption at 880 nm is 7.7 m$^2$/g (Drinovec et al., 2015). Using this value and again assuming the precision in the measurement of absorbance to be $3 \times$

$10^{-6}$, a path length of 3.9 m (390 cm) would be required to obtain a precision of 0.1 µg/m$^3$ for black carbon mass concentration.

Because of the long path lengths required, the pollutants $NO_2$, $SO_2$, and black carbon are difficult to measure by direct absorbance in the gas phase. Several approaches for long path absorption measurements of species in the gas phase have been taken in the past. Open path systems

have used differential optical absorption spectroscopy (DOAS) with path lengths up to many





kilometers (Platt, 1994); however, this limits their use for determining spatial distributions of pollutants. Furthermore, DOAS requires the pollutants detected to have significant structure in the absorption spectrum so that absorptions can be extracted via fitting algorithms.

Closed-path, in situ absorption photometers have typically relied on using mirrors to "fold"

the path length within the detection cell, with up to 100 or more reflections to increase the absorption path length. Of these, the White cell (White, 1942) is the most common. However, even miniaturized versions of White cells have relatively large volumes, typically 180 cm$^3$ and larger, so that the flush times for typical flow rates of 1.8 L min$^{-1}$ are long. Also, the cell shapes required by the mirror arrangements exacerbate the problem, requiring multiple flush times to exchange 99% of the cell

contents (~ 4.6 flush times assuming exponential dilution). Thus, for a cell volume of 180 cm$^3$ (volume of a currently commercially available White cell with 2-meter path length) and flow rate of 1.8 L min$^{-1}$, the total required flush time is 27.6 s. To obtain the low absorbance precisions of $3 \times 10^{-6}$ stated earlier, it is important to measure the reference light intensity ($I_o$) every 5 to 10 seconds due to small intensity fluctuations in typical light sources. This requires total cell flush times of 2.5 to 5

seconds (to measure both $I$ and $I_o$), which is incompatible with White cells unless excessively large (and hence impractical) flow rates are used (> 10 L min$^{-1}$). Other folded-path configurations can be flushed more rapidly (e.g., Herriott cells, Herriott and Schulte, 1965) but require a collimated light source, which is noisier compared to uncollimated sources such as light emitting diodes or low-pressure mercury lamps, thus largely offsetting the advantage in sensitivity gained by the longer path

lengths.

More recent advances employ high-reflectivity cavities to generate long path lengths. Cavity-enhanced absorption spectroscopy (CEAS) and variants such as cavity ring-down spectroscopy (CRDS) and cavity-attenuated phase shift (CAPS) spectroscopy have been successfully used to measure numerous atmospheric constituents in the visible and infrared regions (Paldus and Kachanov,

2004; Crosson, 2008; Kebabian et al., 2005). However, these high-reflectivity cavities are often expensive, and care must be taken such that mirror reflectivity does not degrade over time (resulting in a changing sensitivity and hence a need for frequent re-calibration). Furthermore, they tend to operate over a fairly narrow wavelength range limited by the mirror reflectivities of the cavity.

Here we describe a new approach, a Folded Tubular Photometer, for measurements of a

pollutant or other species in a gas such as air. The use of modular mirror cubes in combination with tubular flow cells allows the path to be folded, making it compact enough for a several-meters-long detection cell to fit into a conventional rack-mount-sized or smaller enclosure that can be produced relatively inexpensively compared to other optical techniques. Further, the design makes it possible to reduce the cell volume and therefore also the flush times significantly, allowing a new $I$ or $I_o$

measurement to be made once every 5 s or less. Because those measurements are made close together in time, variations in the lamp intensity between measurements is small, resulting in higher precision relative to a White cell or Herriott cell of the same path length. Using this approach, measurements of





ambient concentrations of $NO_2$, $SO_2$, and black carbon by direct absorbance in the gas phase become
feasible and economical.

This paper presents a design of the Folded Tubular Photometer that enables rapid
measurements of both $NO_2$ and NO within the same instrument and temporally separated by only a
few seconds. $NO_2$ is measured by direct absorption at 405 nm. NO is measured by addition of ozone
to convert NO to $NO_2$ with nearly 100% conversion by the reaction:

$$NO + O_3 \rightarrow NO_2 + O_2 \tag{3}$$

Subsequent measurement of the increase in $NO_2$ concentration upon addition of ozone provides a
highly accurate measurement of NO. The results described here show that this method provides a
viable approach for measuring both $NO_2$ and NO at atmospheric levels. Alternative commercially
available methods for measuring $NO_2$ based on direct absorbance (CRDS or CAPS) currently measure
$NO_2$ but not NO. Therefore, the FTP described here provides a relatively inexpensive alternative that
measures both $NO_x$ species required for air quality compliance and predictive modeling. The Folded
Tubular Photometer design also will be discussed as it pertains to direct absorbance measurements of
other atmospheric species such as $SO_2$ and BC.

## 2. Experimental

### 2.1 Generalized Folded Tubular Photometer

Fig. 1 is a generalized diagram of the Folded Tubular Photometer for direct measurements via
the Beer-Lambert Law (Eq. (1)) of concentrations of gas-phase molecules by absorption or total
particle extinction (absorption and scattering). An air pump draws sample air through the entire
apparatus. For gas-phase analytes, the sample air enters the instrument through an inert Teflon
particle filter, preventing particles in the sample air from interfering with the absorbance
measurements. The flow then passes through a three-way reference valve, which either directs the air
through a scrubber to remove the analyte from the flowing stream (measuring $I_o$), or through a tube
bypassing the scrubber (measuring $I$). It is desirable that this valve be switched as frequently as
possible to minimize any effect of drift of the lamp intensity between the measurements of $I$ and $I_o$.
However, it is critical to completely flush the detection cell between the $I$ and $I_o$ measurements as well
as allow for adequate signal averaging time of the measured light intensity. This requirement sets a
limit on how frequently the reference valve can be switched. For example, this valve is switched
every five seconds for measuring $NO_2$ for a cell volume of 37.4 $cm^3$ (0.476 cm id, 210 cm long) and
flow rate of 1.8 L/min (30 $cm^3$/s) achieved in our optical bench (described below). This allows for 2
complete flushes of the cell volume within the initial 3 seconds followed by averaging of the light
intensity for the final 2 seconds.

Sample air next passes through one or more parallel tubes composed of Nafion™. Nafion
membranes selectively transport water molecules across the tube wall and bring the humidity inside



the tube to approximately the same level as in the surrounding air. Wilson and Birks (2006) first demonstrated for ozone monitors that small changes in humidity during ozone-scrubbed ($I$) and unscrubbed ($I_o$) measurements resulted in light transmission changes through the optical cell due to adsorption of differing amounts of water vapor on the cell wall. They further showed that use of a

5       Nafion tube just prior to entering the detection cell eliminated this water vapor interference by equilibrating humidity between the $I$ and $I_o$ cycles. Although Nafion can be used to dry the sample (e.g., if the surrounding air has been dried), it is only necessary to equilibrate the water vapor level with the surrounding air to provide equal humidity during both measurement cycles. This has the advantage of not altering the mixing ratio of an analyte by removal of atmospheric water vapor.   For

the examples given here where the typical flow rate is 1.8 L/min, four 25-cm long, 1.07-mm i.d., 1.35-mm o.d. tubes of Nafion (total of 1 m length) plumbed in parallel were found to effectively remove any interference from rapid changes in relative humidity of sampled air. Use of higher flow rates require proportionally larger internal surface areas (longer Nafion tubes at constant i.d.) to prevent humidity interferences. It should be noted that the use of Nafion tubing is not required for particle

measurements since the analyte scrubber can be a hydrophobic particle filter of very low surface area, which absorbs/desorbs very little water vapor. Also, Nafion tubing may cause losses of particles, thereby biasing measurements.

        The air flow next enters the optical bench, which is composed of one or more tubular detection cells (six shown in Fig. 1) and an appropriate number of mirror modules (five shown in Fig.

1), each containing two mirrors oriented at 45° to the flow path. The mirror modules allow sample air to flow through them and to enter the subsequent detection cell. The mirrors direct the light along the same path as the air flow (either in the same or opposite direction – shown in Fig. 1 as counter to the air flow). The mirrors fold the optical path so as to increase the path length and, thus, the sensitivity of the measurement.

The light source module contains a light source that emits light of the appropriate wavelength(s) to be selectively absorbed by the analyte of interest. The preferred light source for most analytes is a light emitting diode (LED), although other light sources may be used. LEDs are readily available with emissions ranging from about 250 nm in the UV to about 1000 nm in the infrared, and have directional light emission that can easily be coupled into the cell. We found LEDs

with bandwidths of a few tens of nanometers to be preferred over laser diodes. Although laser diodes are much brighter, are highly collimated, and have a very narrow bandwidth, they typically exhibit much lower stability (larger fluctuations in intensity on times scales of a few seconds).  In the application described here, an LED with emission maximum at 405 nm was utilized to measure $NO_2$. In other work, we measured black carbon using an LED with maximum emission near 880 nm.

Multiple LEDs may be combined, using either dichroic mirrors or fiber optics, and the LEDs switched on and off to measure multiple species (e.g., $SO_2$ at 290 nm and $NO_2$ at 405 nm in the same air





sample) or to characterize aerosol light extinction over a large wavelength range to characterize particulate composition.

At the end of the optical bench, the light is detected by a photodiode. Typically a large fraction of the light (>90%) from the LED source is lost to partial reflection at the cell walls and
mirrors, and the fraction of light arriving at the photodiode depends on a number of factors such as the intensity and degree of collimation of the light source, reflectivity of the cell walls and mirrors, humidity of the sample, and the pressure inside the detection cell. These losses have no effect on the measurement of the analyte concentration so long as they remain constant during measurements of $I_o$ (analyte scrubbed) and $I$ (analyte present). However, these losses do place a limit on the overall path
length that is achievable at a given wavelength.

The concentration of the analyte (typically in units of molec cm$^{-3}$ for gases) is calculated from the Beer-Lambert Law (Eq. (1)) from the absorption cross section averaged over the bandwidth of the light source; the path length of the light beam, calculated from the dimensions of the optical bench; and the electrical signals (current or voltage) of the photodiode, which are proportional to $I_o$ and $I$.
Since $I_o$ and $I$ are not measured at exactly the same time (typically 5 s apart), one can average the values of $I_o$ measured before and after the measurement of $I$ in order to increase the precision and accuracy of the measurement. Temperature and pressure are measured within the detection cell for the purpose of calculating a mixing ratio of the analyte in typical units of ppm or ppb.

The voltage sensitive orifice (VSO) valve of Fig. 1 serves a particularly important role. It is
used to admit air to the flowing stream after the optical bench and prior to the air pump. Adding air at this point both reduces the flow rate through the optical bench and increases the average pressure. Because the analyte scrubber is more restrictive than the bypass, the pressure within the detection cells is lower when the air is being drawn through scrubber ($I_o$ being measured). To compensate, the VSO valve is adjusted in a feedback loop to increase the cell pressure. The VSO valve is adjusted to
equalize the pressure of the sample air within the optical bench during $I$ and $I_o$ measurements to within an error of 0.1 mbar. This eliminates a potentially large error resulting from the effect of pressure on the transmission of light through the optical bench, which is discussed in Sect. 3.1 below. The flow rate during the $I_o$ measurement is also reduced, but only by ≤ 5% and does not significantly impact the degree of cell flushing. Pressure adjustment is made during the first 2 seconds of the 5-s
cycle, during which the optical cell is also being flushed. The values of $I$ and $I_o$ are measured in the final 2 seconds of the corresponding 5-s cycles after the pressure adjustment is achieved and the cell has been thoroughly flushed.

## 2.2    Modular optical bench

A more detailed perspective drawing of the modular optical bench, as used in the work described here, is shown in Fig. 2. Six tubular detection cells and 5 mirror modules are shown, although other numbers of tubular detection cells and mirror modules could be used. In this example,



not all of the detection cells are of the same length, so as to make room on the optical bench for both the LED light source and the photodiode detector. Tubing connections for the air inlet and air outlet are shown. The flow could be reversed with no effect on the analyte measurement. Each mirror module contains two mirrors. The optical bench constructed for use in the examples that follow made

use of o-rings to seal the two ends of the tubular detection cells to the mirror modules, LED module, and photodiode module. The mirror, light source, and light detector modules are mounted to a vibrationally-isolated, rigid optical bench. The modular nature of the optical bench allows the path length to be increased or decreased by adding or removing tubular cells and mirror modules as desired for measurements of analytes in varying absorbance ranges. Also, as shown in Fig. 2, tubular cells

may be of different lengths, making a wide range of path lengths possible. The materials used for construction of the detection cells should be inert toward the analyte being measured, with no significant loss of the analyte to exposed surfaces. The examples given below made use of an optical bench constructed of aluminum. To increase transmission of light, the interiors of the cell were polished using either a cylinder hone or a metal bristle brush of the type used to clean gun barrels.

For the $NO_2$ photometer discussed below, we used tubular cells with 3/16-in (0.476 cm) i.d. such that a 2.1-m long absorption cell has a calculated volume of only ~37.4 cm$^3$. Thus, the time for one flush at a flow rate of 1.8 L/min (30 cm$^3$/s) is only 1.25 s. The time for a molecule to diffuse across the inner diameter of the cell is calculated to be ~0.5 s, and as a result, nearly plug flow results and only one or two flush times are required to achieve greater than 99% complete flushing of the

previous contents of the cell. This allows a new $I$ or $I_o$ measurement to be made once every 5 s or less, thereby reducing variations in the lamp intensity between measurements. As a result, the precision achieved is higher than is possible in a White or Herriot cell of the same path length.

### 2.3 Folded Tubular Photometer for measurements of NO₂ and NO

Fig. 3 is a schematic diagram of the inlet system of a Folded Tubular Photometer designed to measure both nitrogen dioxide ($NO_2$) and nitric oxide (NO) using a LED light source with maximum emission at 405 nm. $NO_2$ absorbs at 405 nm with an absorption cross section of ~$6 \times 10^{-19}$ cm$^2$molec$^{-1}$ (Burrows et al., 1998). NO does not absorb at this wavelength but can be quantitatively converted to $NO_2$. The reference $NO_x$ scrubber contains a combination of manganese dioxide to oxidize NO to

$NO_2$ followed by activated carbon to remove $NO_2$. The entire scrubber is heated to 110 °C. The inlet system is the same as in Fig. 1 but with some additions (shown in the gray box) that allow conversion of NO to $NO_2$ by the highly selective reaction of NO with $O_3$ (reaction (3)). This is accomplished by adding a small flow (< 5% of total instrument flow) of ozonized air (produced photolytically by a low-pressure mercury discharge lamp) and allowing them to react within a reaction coil during a third

measurement step. The three measurements steps are shown in the panels of Fig. 3. In Fig. 3a (air flow paths shown in red), $I_o$ for $NO_2$ is measured as the sample air passes through the $NO_x$ scrubber, removing both NO and $NO_2$. In the shaded gray box, approximately ~70 cm$^3$/min of air, scrubbed of





both ozone and NO$_x$, bypasses the ozone generator and is added to the sample air stream. Correction is made in the firmware for dilution of NO$_2$ and NO in the air sample by this small flow. The bypass valve then directs the combined flow to bypass the reaction coil, pass through the Nafion humidity equilibrator and enter the optical bench.

5         Fig. 3b (air flow paths shown in green) differs from Fig. 3a only in that the state of the reference valve is changed so that sample air bypasses the reference NO$_2$ scrubber. The NO$_2$ present in the sample stream now attenuates light passing through the optical bench, and the light intensity $I$ is measured for NO$_2$. Using the value of $I_o$ measured in configuration 3a and $I$ measured in configuration 3b, the NO$_2$ concentration can now be calculated using the Beer-Lambert Law, Eq. (1).

The light intensity measured using configuration (b) also serves as the $I_o$ for calculation of the NO concentration.

        In Fig. 3c (air flow paths shown in blue), the states of both the ozone and bypass valves are changed such that the small flow ($\sim$ 70 cm$^3$ min$^{-1}$) passes through the photolytic ozone generator, and the ozonized air mixes with the sample air and passes through the reaction coil where NO reacts with

ozone to form NO$_2$. The ozone mixing ratio in the combined streams (ozonized air mixed with sample air) is typically 8 ppm. The reaction coil is constructed from a 1-m coiled length of 0.635-cm i.d. PTFE, producing a reaction volume of 31.7 cm$^3$ and residence time for a total flow rate of 1.8 L min$^{-1}$ of 1.06 s. Based on the reaction rate coefficient of k$_3$ = 1.9 $\times$ 10$^{-14}$ cm$^3$ molec$^{-1}$ s$^{-1}$ at 298 K (Birks et al., 1976; Borders and Birks, 1982; Burkholder et al., 2015) and a total pressure of 1 atm, the

reaction is calculated to be 97.6% complete within the reaction coil. Nearly all of the remaining 2.4% of NO is converted during transit through the optical bench. Assuming pseudo-first order kinetics, the average amount of converted NO detected within the optical bench and measured is calculated to be 98.8%. It should also be noted that the combined residence time within both the reaction coil and the detection cells is ~2.2 s, which allows for a complete flush of the detection volume prior to

measuring the light intensity.

        The light intensity measured using configuration (c) serves as the value of $I$ in the calculation of NO using Eq. (1). For NO measurements, correction for incomplete reaction may be made by dividing by the average of the fraction of NO converted; i.e. 0.988 for the flow conditions described above. In practice, air standards having known NO and NO$_2$ concentrations were used to calibrate the

outputs of the instrument to correct for incomplete reaction and any other factors affecting the sensitivity and offset of the instrument.

        To summarize (and shown schematically in Fig 3d), by continuously cycling between valve states (a), (b), and (c) every 5 seconds, a new value of either NO and/or NO$_2$ may be calculated and updated as follows: (a) a new value of $I_o$ for NO$_2$ is measured, allowing calculation and updating of

a new value of NO$_2$ concentration, (b) a new value for both $I$ for NO$_2$ and $I_o$ for NO are measured, allowing calculation and updating of new values of NO$_2$ and NO, and (c) a new value of $I$ for NO is measured, allowing calculation and updating of a new value of NO. It should also be noted that if



only NO$_2$ measurements are desired, step (c) can be omitted (and the small flow that delivers ozone discontinued). Conversely, step (a) can be omitted if only NO measurements are desired.

## 3. Results and discussion

### 3.1 Effect of pressure on analyte measurements

A problem we encountered when attempting to use long tubular detection cells, with the light beam either folded using mirrors or unfolded, is that the transmission of light through the cell was found to be pressure dependent. For example, the pressure difference resulting from flowing a sample gas directly into the cell during the measurement of $I$ vs. flowing through the solid-phase NO$_2$ scrubber during the measurement of $I_o$ at a flow rate of ~1.8 L min$^{-1}$ was found typically to be ~10 mbar. This pressure difference alone causes an unacceptable offset error of typically ~50 ppb in the measurement of NO$_2$. Although correction can be made for the offset, the offset may change due to variations in the conductance of the scrubber, which varies with environmental factors such as humidity, thereby introducing unacceptable levels of low frequency noise (drift).

Fig. 4 illustrates the observed pressure variation during measurements of $I_o$ and $I$ on the measurement of NO$_2$ concentration. In this plot, pressure difference is the pressure of the cell during the $I$ measurement (Fig. 3b) minus the pressure in the cell during the $I_o$ measurement (Fig. 3a). Since the scrubber adds to the restriction, the pressure is typically lower during the $I_o$ measurement. To enable adjustments of the pressure difference, a needle valve was placed in line with the analyte scrubber or in line with the bypass and the restriction was varied. Results are provided in Fig. 4 for two prototype NO$_2$ monitors constructed. The presence of unmatched pressures during the $I_o$ and $I$ measurements was found to produce a false reading, or offset, that is additive to the true NO$_2$ concentration. As can be seen in Fig. 4, the offset varies linearly over the range tested ($\Delta P$ = -9 to +13 mbar) and can be quite large, ranging from -100 to +150 ppb. The slopes of the regression lines for the two prototypes differ, ranging from 5.0 to 12.5 ppb/mbar, and we found that such slopes vary from instrument to instrument. As discussed below, we believe that this offset is due to changes in the transmission of light through the optical bench with change in pressure, mostly likely because of the effect of pressure on the refractive index of the sample gas, but possibly due to other factors.

The magnitude of the pressure dependence on light transmission is unexpected and not easily explained by any existing theory. For example, it cannot be accounted for by differences in Rayleigh scattering by air molecules at different densities. The Rayleigh scattering cross section in air is ~10$^{-27}$ cm$^2$ molec$^{-1}$ at 532 nm. For a path length of 210 cm and temperature of 25 °C, a 10-mbar pressure change would cause an extinction change of only ~5 × 10$^{-8}$, nearly two orders of magnitude below the limit of detection for our absorbance measurements.

The effect may be due to variations in the propagation of the non-collimated beam of light through the cell by reflection from the cell's internal surface and/or mirrors used to fold the path, a



process that depends on the refractive indices of the sample gas (which depends on pressure) and the cell wall or mirror surface. The effect also could be due to changes in the amount of adsorbed water vapor on the interior surface of the cell that is in equilibrium with water vapor in the gas phase, the concentration of which changes with cell pressure (and therefore density), or it might be due to subtle

changes in the optical alignment with pressure change.

The effect of pressure on absorbance measurements was made insignificant by controlling the pressure during measurements of $I_o$ and $I$ to be identical to within 0.1 mbar, using the VSO valve shown in Fig. 1. This degree of pressure control yields offsets in the range 0.5 to 1.25 ppb for the two prototype instruments evaluated for pressure effects. Such small offsets are easily removed by

applying an additive offset calibration factor determined while passing the sample air through a zeroing $NO_x$ scrubber.

## 3.2 Analytical figures of merit for NO and NO₂

The Folded Tubular Photometer configured for measurements of $NO_2$ and NO (Fig. 3) is now

commercially available from 2B Technologies (Boulder, CO) as the Model 405 nm $NO_2$/NO/$NO_x$ Monitor™. It was externally tested and approved as a Federal Equivalent Method (FEM) for monitoring of the Criteria Pollutant $NO_2$ for compliance with the U.S. Clean Air Act (designated as EQNA-0217-243). During the period 1 April 2016 through 30 November 2017, 206 calibrations were performed on 41 different instruments. Calibration curves were constructed at five

concentrations (0, 50, 100, 150, and 200 ppb) for both $NO_2$ and NO. Standard concentrations of $NO_2$ and NO were generated using an API Model 700 Dynamic Dilution Calibrator. An internal photolytic ozone source and photometer generates known concentrations of ozone, which react with an excess of NO supplied by a compressed gas cylinder to produce a stoichiometric increase in $NO_2$ and decrease in NO concentration, according to reaction (3) above. The internal ozone photometer is traceable to

NIST through a Thermo Electron Model 49i-PS Ozone Calibration Primary Standard. Typically, five independent calibrations were carried out for each instrument and linear regressions applied to each calibration curve. The instruments were found to be highly linear over this concentration range with coefficients of determination $R^2$ averaging 0.9995 and 0.9993 for $NO_2$ and NO, respectively, for the 206 calibrations performed. Although typical calibrations only cover the range of 0-200 ppb for

ambient measurements, strict linearity up to 1000 ppb has been observed and the linear dynamic range is estimated to extend to 10,000 ppb (10 ppm) for $NO_2$ and 2,000 ppb (2 ppm) for NO. The dynamic range for NO is limited by the ozone concentration (~8 ppm) used to convert NO to $NO_2$.

Precisions (1σ) obtained in dual mode (both $NO_2$ and NO measured) for 5-second measurements were typically in the range 2-3 ppb with an average of 2.3 ppb. When operating in

single mode (only NO or $NO_2$), the response time is 10 s, the time required to obtain a new measurement of both $I$ and $I_o$. In dual mode, the response time is increased to 15 s (one of the measurement cycles simultaneously provides $I$ for $NO_2$ and $I_o$ for NO (Fig. 3), thus shortening the





response time from what would otherwise be 20 s). Averaging can be used to trade off response time for improved precision. Ambient air monitors commonly employ a conditional averaging filter for improving the signal-to-noise ratio of this measurement. This consists of maintaining both a short-time running average (~20-30 sec) and a long-time running average (~2-5 minutes). When measured

concentrations are stable, the long-term average is output; however, when rapid concentration changes occur, the short-term average is output. This type of filtering has the advantage of providing improved precision while maintaining the ability to respond relatively fast to large concentration changes. The averaging times and threshold concentration changes of the conditional averaging filter are user selectable in the Model 405 nm monitor. For averaging times of 3 minutes, the precisions

were found to be independent of test concentration over the 0-200 ppb calibration range, averaging 0.386 ppb for $NO_2$ and 0.381 ppb for NO for the 206 calibrations performed. Other specifications for the Model 405 nm $NO_2$/NO/$NO_x$ Monitor that include physical and electrical parameters like size, weight and power requirements in addition to figures of merit are provided in Table 1.

**3.3 Interferences in the measurement of $NO_2$ and NO**

Interferences in the UV-absorbance technique occur when either (1) other species that absorb the same wavelengths of light as the analyte are also removed by the scrubber or (2) species that can somehow affect light transmission (such as the aforementioned water vapor interference in ozone monitors) are altered by passing through the scrubber. In considering the magnitude of possible

interferences, one must consider both the ambient concentration and the absorption cross section (for Eq. (1)) of the interfering species as well as whether it is removed or significantly interacts with the scrubber. As part of the requirements for FEM designation, the Model 405 nm $NO_2$/NO/$NO_x$ Monitor was tested for interferences from high concentrations of common atmospheric constituents. Carbon monoxide, nitric oxide, ozone, sulfur dioxide and water vapor were added in the presence of

100 ppb of $NO_2$. Ammonia was also tested without $NO_2$ present. None of these compounds exhibit absorbance at 405 nm, but they can have large enough ambient concentrations to possibly influence light transmission in the detection cells. However, all measured responses were insignificant within statistical error, the highest response being $1.2 \pm 2.3$ ppb from 50.7 ppm of CO. Increasing the relative humidity from dry (RH < 1%) to ~20,000 ppm of water vapor (55% RH at 24.8 °C) gave an

insignificant response of $0.3 \pm 2.4$ ppb. The results of interference testing are summarized in Table 2.

$NO_2$ absorption at 405 nm is particularly attractive because there are almost no airborne species that absorb significantly at this wavelength other than $NO_2$. Aromatic compounds (which can present interferences for ozone at 254 nm, Turnipseed et al., 2017) do not show significant absorption above 300 nm (Kelly-Rudek et al., 2013). Only multiple-ringed aromatics are known to have

significant absorption near 405 nm and their gas-phase concentrations are exceedingly low (few ppt) due to their low vapor pressures. These compounds tend to partition to the aerosol phase (Finlayson-



Pitts and Pitts, 2000), and particulates (along with any extinction due to particulates) are removed by the inlet Teflon particle filter of the instrument (Figs. 1, 3). HONO, $NO_3$, glyoxal and methyl glyoxal exhibit absorption near 405 nm (see Fig. 5), but the cross sections of these compounds are $\geq 6$ times less than $NO_2$. $NO_3$ is highly reactive and is only present at low ppt levels at night near the earth's

surface (Brown et al., 2007). Stutz et al. (2004) report that ratios of [HONO]/[$NO_2$] in urban areas reach maximum values of only 0.1 at night with concentrations of only a few ppb at most (Bernard et al., 2016). Kebabian et al. (2008) report a minor interference from glyoxal and methylglyoxal in their CAPS $NO_2$ monitor during measurements in Mexico City; however, the CAPS operated at a wavelength of 440 nm (with a $\pm$ 10 nm bandwidth) where the absorption cross section of both the

glyoxal and methylglyoxal is considerably larger (see Fig. 5). Both of these compounds also have only been observed to be at most a few ppb even in polluted urban atmospheres (Vrekoussis et al., 2009). At typical concentration levels, interferences from all of these possible $NO_2$ interferences are expected to be negligible at 405 nm.

    As mentioned previously, NO is measured by quantitatively converting it to $NO_2$ by reaction

with excess ozone (reaction (3)). Although this is a simple bimolecular reaction with a known $NO_2$ yield of unity, subsequent chemistry could affect $NO_2$ concentrations within the photometer. Specifically, the large excess of ozone used (~ 8 ppm) can also slowly convert $NO_2$ to $N_2O_5$ via:

$$NO_2 + O_3 \rightarrow NO_3 + O_2 \tag{4}$$
$$NO_3 + NO_2 \rightleftharpoons N_2O_5 \tag{5}$$

Reaction (4) is ~600 times slower than reaction (3) ($k_4 = 3.22 \times 10^{-17}$ cm$^3$ molec$^{-1}$ s$^{-1}$ at 298 K, Burkholder et al., 2015), yet can proceed to a small extent at high $NO_x$ levels. At room temperature and $NO_2$ concentrations greater than about 25 ppb, the reaction (5) equilibrium favors $N_2O_5$ formation and proceeds relatively rapidly ($k_5 = 1.4 \times 10^{-12}$ cm$^3$ molec$^{-2}$ s$^{-1}$, Burkholder et al., 2015), thus removing $NO_3$ and resulting in a net loss of 2 $NO_2$ molecules. Loss of $NO_2$ within the reaction coil

and detection cells due to reactions (4) and (5) will result in a slight increase in light transmission, thereby causing an underestimate of the NO concentration. Evidence for this chemistry was observed by adding $NO_2$ to the analyzer in the absence of NO (Fig. 6). As seen in the figure, measured NO mixing ratios apparently decrease with increasing $NO_2$. The linear fit of the data gives a slope of -3.4 ppb NO/100 ppb of $NO_2$. Assuming that the reaction of $NO_3$ with $NO_2$ is fast and that the NO + $O_3$

reaction goes to near completion before significant $NO_2$ is lost via reaction (4) (valid since $k_3/k_4$ ~600), a simple correction can be derived from assuming pseudo-first order kinetics, along with an estimate of [$O_3$] from the photolytic generator, measured [$NO_x$], the cell temperature, and cell flow rate:

$$[NO]_{corr} = 2[NO_x]_o(1 - e^{-k_4[O_3]t}) \tag{6}$$

Here, [NO]$_{corr}$ is the amount of $NO_2$ that is lost due to reactions (4) and (5) and should be added to the measured NO concentration. The residence time ($t$) in the reaction coil and optical cells is computed





from the effective reaction volume (reaction coil volume plus half the optical cell volume) and the measured flow rate; $k_4$ is the temperature-dependent rate coefficient of reaction (4); and the factor of 2 results from the stoichiometry of reactions (4) and (5). $[O_3]$ has been observed to vary between 8 and 10 ppm for the typical flow rates and photolysis lamp intensities (which are measured) used in the

analyzer. $[NO_x]_o$ is estimated as the sum of the most recently measured $NO_2$ concentration and the most recently measured uncorrected NO concentration. Using the uncorrected NO concentration to compute $[NO_x]_o$ does cause a slight underestimation in the correction (since NO is underestimated at this point). This underestimation can be eliminated by applying this correction in an iterative fashion – computing a corrected NO, then using this to recompute the $[NO_x]_o$. However, it was found that use

of a single iteration resulted in corrections that were within the instrumental measurement precision of those that used only the uncorrected NO. As can be seen in Fig. 6, use of Eq. (6) over the range of expected $O_3$ mixing ratios reduces the observed NO bias by a factor of three (to ~ -1 ppb NO/100 ppb $NO_2$). That the bias is not completely removed may be due to reaction (4) being slightly faster than reported, or, more likely, a heterogeneous contribution to reaction 4 on the optical cell walls. For

ambient levels of NO and $NO_2$, the measurement error for NO is well within the noise of the instrument after applying this correction in the firmware.

### 3.4  Roadside measurements of $NO_2$ and NO

        The Folded Tubular Photometer (Model 405 nm) was tested for $NO_2$ and NO at a roadside

monitoring site operated by the Colorado Department of Public Health and Environment (CDPHE) in the fall of 2015 for five days. The monitoring station was located along an entrance ramp at the intersection of Interstate 25 and Interstate 70 just north of downtown Denver (I-25/Globeville site; see https://www.colorado.gov/airquality for site details). Air was sampled through a Teflon inlet line that was located within 1 m of the road at an approximate height of 4 m. Air was drawn into the

instrument shelter and sampled into three analyzers: (1) the 2B Model 405 nm $NO_2/NO/NO_x$ Folded Tubular Photometer, (2) a Teledyne-API Model T500U cavity-attenuated phase shift spectrometer (CAPS) for $NO_2$ and (3) a Teledyne-API Model 200E $NO/NO_x$ chemiluminescence (CL) monitor that reported both NO and $NO_2$. Both the T500U and the 200E have either FEM or FRM (Federal Reference Method) designation and are operated by CDPHE as part of the State of Colorado's

ambient air quality monitoring network. The Model 405 nm Folded Tubular Photometer was operated at a flow rate of 1.6 L min$^{-1}$ in both a dual mode for both $NO_2$ and NO for the first 4 days and then in a single $NO_2$-only mode for one additional day. Unfiltered 5-second data from the Model 405 nm was logged and then-averaged to 1 minute for comparison with the reference analyzers for $NO_2$ and NO.

        Fig. 7 shows the time series for 1-minute averaged measurements of $NO_2$ for both the Model

405 nm Folded Tubular Photometer and the Teledyne-API CAPS. Note that the $NO_2$ data for the Folded Tubular Photometer plotted in Fig. 7 are shifted by 50 ppb for comparison purposes. The agreement between the two data series is excellent, with both data sets capturing the same sharp





changes in concentration due to rapidly changing concentrations in the roadside environment. Also shown is the correlation plot for the data. The linear regression line of this plot has a slope of 0.968, an intercept of -0.5 ppb, and a coefficient of correlation ($R^2$) of 0.960.

5         Fig. 8 compares the 1-minute averaged measurements of NO mixing ratio using the Model 405 nm Folded Tubular Photometer with simultaneous 1-minute averaged measurements made by the Teledyne API Model 200E Chemiluminescence Analyzer (CL). The method used by this analyzer, detection of chemiluminescence in the reaction of NO with a large excess of ozone (Fontijn et al., 1970), is the most common method used for ambient air measurements of nitric oxide. Data for the Folded Tubular Photometer are shifted in Fig. 8 by addition of 400 ppb for clarity. NO mixing ratios

were not initially corrected for $N_2O_5$ formation (see Sect. 3.3). Yet the agreement between the two measurement methods is excellent even when NO mixing ratios were changing rapidly. The data are also shown as a correlation plot. The linear regression yields a slope of 0.947, an intercept of -0.2 ppb, and a coefficient of correlation ($R^2$) of 0.976. Correction of the data for $N_2O_5$ formation (Eq. (6)) yielded a slope of 0.973 and an intercept of 0.3 ppb (the correlation coefficient remained the same,

0.976), showing the small, but significant magnitude of the $N_2O_5$ correction over this $NO_x$ concentration range. Note that once corrected, the correlation slope for NO is identical to that of $NO_2$. The slight deviation from unity for these slopes is likely due to differences in calibration factors and possibly due to small timing offsets caused by the slightly different inlet plumbing between the instruments in a roadside sampling environment where concentrations were highly variable and

changing rapidly.

## 4. Conclusions and future directions

We have developed and characterized a new instrumental method, the Folded Tubular Photometer, for

measurements of ambient concentrations of both $NO_2$ and NO in air. The instrument is commercially available as the Model 405 nm $NO_2/NO/NO_x$ Monitor (2B Technologies, Inc., Boulder, CO). A design using a folded tubular optical bench yields long path lengths with low cell volumes, thereby enabling $NO_2$ to be measured directly via optical absorbance at 405 nm. This is sometimes referred to as "true $NO_2$" and is essentially interference free. The instrument measures NO by conversion to $NO_2$

via the addition of ozone, thus enabling the full characterization of $NO_x$ (NO + $NO_2$) needed for photochemical modeling. The optical bench is modular and can have variable path lengths of up to 2 m or longer. The cell design makes it possible to measure other species that are typically difficult to measure by direct absorption photometry, such as $SO_2$ and black carbon. Pressure equalization during the various stages of the absorbance measurement cycle is critical to obtaining accurate measurements

of the analyte.

        The most common method to measure $NO_2$ has long been reduction to NO, followed by chemiluminescence with ozone (Fontijn et al., 1970). This indirect technique has several





disadvantages. The most common means of $NO_2$ reduction is involves passing the air sample through
a heated molybdenum catalyst bed (Winer et al., 1974). However, it has been well established that
other nitrogen species in the atmosphere, especially peroxyacetyl nitrates (PANs), $N_2O_5$, and nitric
acid ($HNO_3$), may be converted as well (Winer et al., 1974; Dunlea et al., 2007). It is often observed

that the conversion efficiency for $NO_2$ is not unity or changes with extended use as required by long-
term monitoring. Photolytic reduction of $NO_2$ to NO is more selective (Parrish et al., 1990; Parrish
and Fehsenfeld, 2000; Buhr, 2007), yet the photolytic conversion is often much less than unity
(typically ~50%). Furthermore, a photochemical equilibrium is established within the photoreactor
between NO, $NO_2$, and $O_3$ resulting in a dependency of the conversion efficiency on the ambient

concentration of ozone (Parrish et al., 1990). Recent work in extremely polluted environments
(tunnels with high vehicle traffic) also shows evidence for undesired photochemistry from
hydrocarbons that biases the $NO_2$ measurements (Villena et al., 2012). Furthermore, regardless of the
conversion method, the measurement of $NO_2$ is indirect, being calculated from the difference between
measurements of $NO_x$ ($NO_2$ + NO) obtained by passing through the converter and measurements of

NO without passing through the converter.

        Direct measurement of $NO_2$ by absorption photometry, such as in the Folded Tubular
Photometer described here, avoids the problems of this long-established indirect chemiluminescence
method. Cavity ring-down spectroscopy (CRDS) and cavity-attenuated phase-shift (CAPS)
spectroscopy are also direct $NO_2$ measurement techniques based on light absorption at 405 and 450

nm, respectively, and are also commercially available (Kebabian et al., 2008; Beaver et al., 2013).
These systems use a high-finesse optical cavity to reflect the optical light beam multiple times to
generate very long path lengths, thus increasing the sensitivity. However, the cost of both the CAPS
and CRDS instruments are significantly higher than the single-pass Folded Tubular Photometer
absorbance analyzer described here, primarily due to the expense of the high-finesse cavities and

associated optics. Our intercomparisons of $NO_2$ measurements made by the Folded Tubular
Photometer and a CAPS $NO_2$ analyzer in a highly trafficked roadside environment showed excellent
agreement for concentrations up to ~ 85 ppb (as large of a range as one would expect for ambient
measurements). Thus, the Folded Tubular Photometer can achieve comparable measurement
accuracy for air quality monitoring at significantly less cost.

30        Furthermore, the commercially-available CRDS and CAPS instruments do not measure NO.
Since NO and $NO_2$ are in rapid photochemical equilibrium, measurements of both are required to fully
characterize either ambient concentrations or emissions from industrial sources. Certainly both are
needed as inputs to regional chemical-transport models that predict air quality. As demonstrated here,
the low cell volume allows the Folded Tubular Photometer method to convert NO to $NO_2$ via addition

of ozone, thereby enabling accurate measurements of NO in the Model 405 nm Folded Tubular
Photometer. Even though this measurement of NO is indirect in nature, the conversion efficiency is
near unity, and we observed excellent agreement in a roadside intercomparison with the standard NO



chemiluminescence technique for concentrations up to 500 ppb of NO. Slight corrections are
necessary due to $N_2O_5$ formation in the photochemical reaction coil, but these are typically small –
less than 3% in the observed roadside study – and correctible within the firmware by means of a
simple kinetics model.

In summary, compared to other available instruments, the Folded Tubular Photometer method
provides a direct, accurate measure of $NO_2$; also measures NO; is less expensive; and is smaller,
lighter, and consumes less power, making it an attractive alternative for compliance monitoring sites
and field measurements of these important atmospheric species. As with absorption photometers for
ozone, calibration depends primarily on the known path length and absorption cross section and does
not vary in time. Thus the Folded Tubular Photometer provides the robust, accurate measurement of
$NO_2$ and NO that is necessary for long-term compliance monitoring.

As suggested in Sect. 1, the Folded Tubular Photometer may be applied to direct
measurements of other atmospherically significant species including $O_3$, $SO_2$, and optical extinction of
aerosols for characterization of particulates. Sulfur dioxide has typically been measured by
fluorescence (Schwarz et al., 1974). However, absorbance has the advantage of being an absolute
method, requiring only infrequent calibration. Instruments based on absorbance are typically less
expensive to construct than fluorescence-based instruments and require less power because a high
intensity light source is not required. Thus, an instrument based on direct absorbance of $SO_2$ would
have advantages over fluorescence, at least in those applications where it provides adequate
sensitivity. Ambient ozone also is a significant interference for $SO_2$ as it absorbs in the same region
as $SO_2$ (Kelley-Rudek et al., 2013) and is typically present at much higher concentrations. However,
direct absorbance measurement of $SO_2$ could be useful in applications such as smokestack monitoring
for $SO_2$ emissions in the combustion of fossil fuels such as coal or natural gas. Here, concentrations
are relatively large, ozone is absent, and a more robust instrument requiring little maintenance and
infrequent calibration is desirable.

The Folded Tubular Photometer can also be applied to measurements of particulate extinction
(defined as the sum of aerosol light absorption and scattering). Large multipass extinction cells have
been used (e.g., Schnaiter et al., 2005), but lack the necessary precision due to the inability to flush
the large volume cells, as discussed in Sect. 1. Cavity techniques (CRDS and CAPS) have both been
applied successfully to particulate extinction (Moosmüller et al., 2005; Massoli et al., 2010), but the
highly reflective mirrors required in these cavities only operate over a small range of wavelengths
(10-50 nm) (Zhao et al., 2014; Washenfelder et al., 2013). Thus, they are incapable of measuring
across wide spectral ranges (e.g., from UV to the near-IR) without the use of multiple cavities, which
would add significant cost. Understanding the spectral dependence of particulate extinction is often
desired to infer both aerosol size and composition. The mirrors used in the Folded Tubular
Photometer-based analyzer have adequate reflectivity (>90%) from 350 to 1000 nm; therefore,



multiple wavelengths of light from different LED sources can be combined via dichroic mirrors or by fiber optics and passed through the detection cells.

      In polluted urban areas, extinction in the near IR (~880 nm, where light scattering by sub-micron particles is weak) can approximate absorption by black carbon. Although not specifically regulated in the U.S., black carbon has been linked to numerous cardio-respiratory illnesses (U.S.-EPA, 2012). Black carbon has long been measured by the method of aethalometry whereby particulate matter is continuously deposited on a filter and transmission of light through the filter is continuously monitored (Hansen et al., 1982). However, aethalometers have been shown to have several artefacts associated with light scattering by the filter medium, loading corrections, and agglomeration of particulates (Weingartner et al., 2003; Arnott et al., 2005; Coen et al., 2010; Baumgardner et al., 2012). Operation of an extinctiometer at 880 nm based on the Folded Tubular Photometer would provide an accurate estimate of black carbon concentrations in urban areas, free from the artefacts caused by filter collection.

      These examples represent a few possibilities for the Folded Tubular Photometer. We have demonstrated its usefulness in the measurements of $NO_2$ and NO. For other species it has the potential for providing accurate measurements with a robust technique (akin to the long-standing absorbance method of measuring ozone) that needs infrequent calibration and can be produced at lower cost than existing technologies.

**5. Acknowledgements**

Air Resource Specialists of Fort Collins, Colorado provided the external testing for EPA Federal Equivalent Method certification.





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



Table 1. Analytical and Physical Specifications, 2B Technologies Model 405 nm $NO_2$/NO/$NO_x$ Folded Tubular Photometer

| | |
|---|---|
| Linear Dynamic Range | 0-10,000 ppb (0-10 ppm) for $NO_2$; 0-2,000 ppb (0-2 ppm) for NO |
| Measurement Frequency | 0.2 Hz (once every 5 s) |
| Resolution | 0.1 ppb |
| Accuracy | Greater of 2 ppb or 2% of reading |
| Precision ($1\sigma$ rms noise) | < 0.5 ppb (with conditional averaging filter), < 3 ppb (no conditional averaging filter) |
| Limit of Detection ($2\sigma$) | < 1 ppb (with conditional averaging filter) |
| Response Time | 10 s single mode ($NO_2$ or NO measured) 15 s dual mode (both $NO_2$ and NO measured) 20 s (with conditional averaging filter) |
| Flow Rate (nominal) | 1.5 Liter/min |
| Operating Temperature | 10 to 50 °C (FEM approved for 20-30 °C for $NO_2$) |
| Power Requirement | 11-14 V dc or 120/240 V ac, 1.3 A at 12 V, 16 watt |
| Size | Rackmount: 17" w × 14.5" d × 5.5" h (43 × 37 × 14 cm) |
| Weight | 18.6 lb (8.4 kg) |





Table 2. Interference Test Results for FEM Certification, 2B Technologies Model 405 nm
$NO_2/NO/NO_x$ Folded Tubular Photometer

| Interferent | $NO_2$ Concentration, ppb | Interferent Concentration, ppb[1] | Response to Interferent, ppb |
|---|---|---|---|
| $NH_3$ | 0 | 100.9 | -0.5 ± 1.2 |
| CO | 99.1 | 50,700 | 1.2 ± 2.3 |
| NO | 99.8 | 509.6 | 0.6 ± 1.9 |
| $O_3$ | 101.2 | 500.0 | 0.2 ± 3.9 |
| $SO_2$ | 100.9 | 519.7 | 0.1 ± 1.9 |
| $H_2O$ | 101.8 | 20,150 ppm | 0.3 ± 2.4 |

[1]Units for water vapor are ppm as noted.





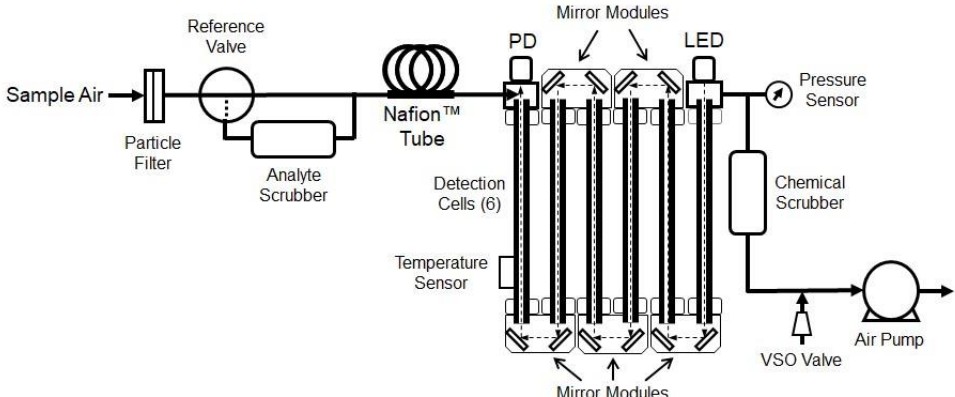

**Figure 1.** Schematic diagram of a Folded Tubular Photometer for measuring the concentrations of
gas-phase species such as $O_3$, $NO_2$, and $SO_2$, and particulates such as black carbon, based on the
absorbance of UV ($O_3$, $SO_2$), visible ($NO_2$, black carbon), or infrared (black carbon) light.





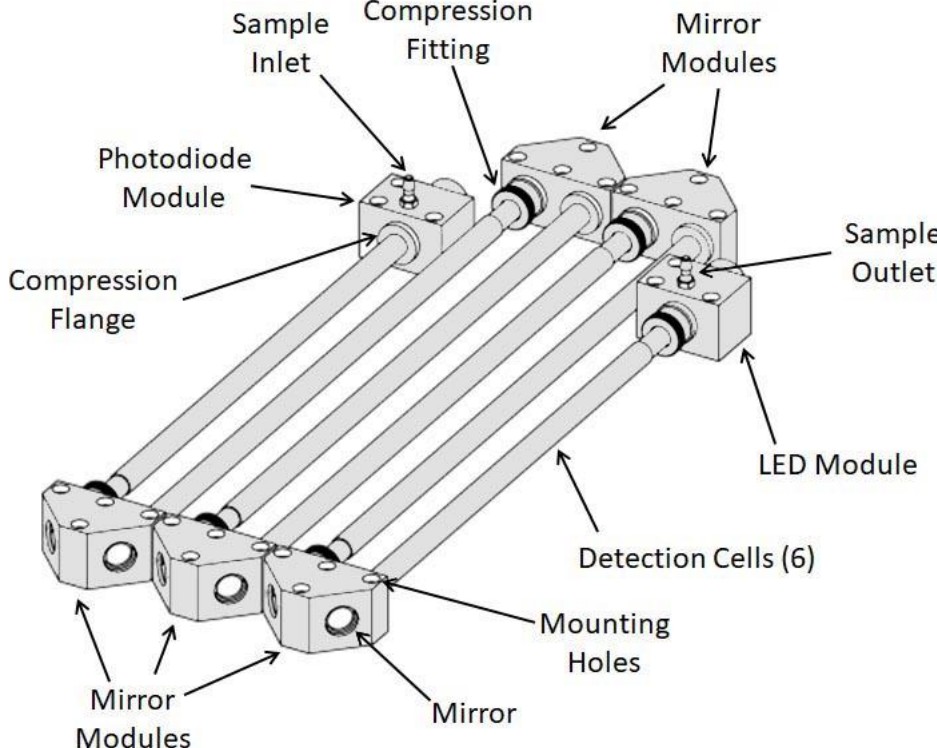

**Figure 2.** Perspective drawing showing the various modular components of a Folded Tubular Photometer.





**a. Measure  $I_o$  for NO$_2$**

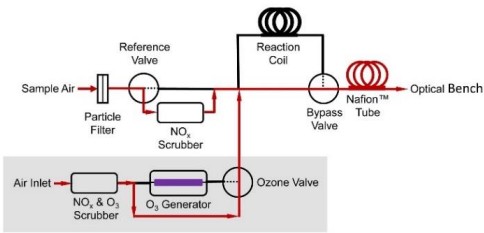

**b. Measure  $I$ for NO$_2$ and $I_o$ for NO**

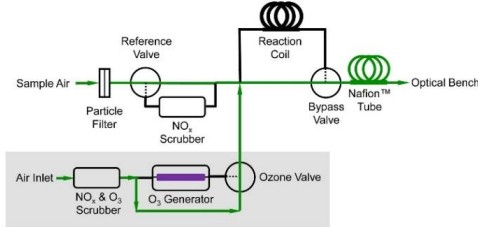

**c. Measure  $I$ for NO**

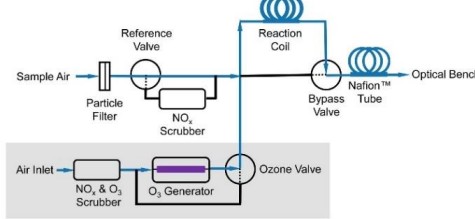

**d. Timing**

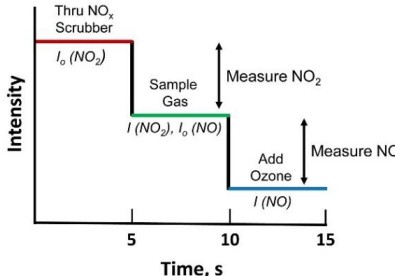

**Figure 3.**  Schematic diagram showing the 3-way valve states for measuring (a) $I_o$ for NO$_2$; (b) $I$ for
NO$_2$ and $I_o$ for NO; and (c) $I$ for NO.  Flow path is shown in red, green and blue for panels (a), (b) and
(c), respectively.  Panel (d) depicts an idealized measurement sequence corresponding to the 3 steps
shown in (a)-(c) and indicates the timing of the three measurement stages.




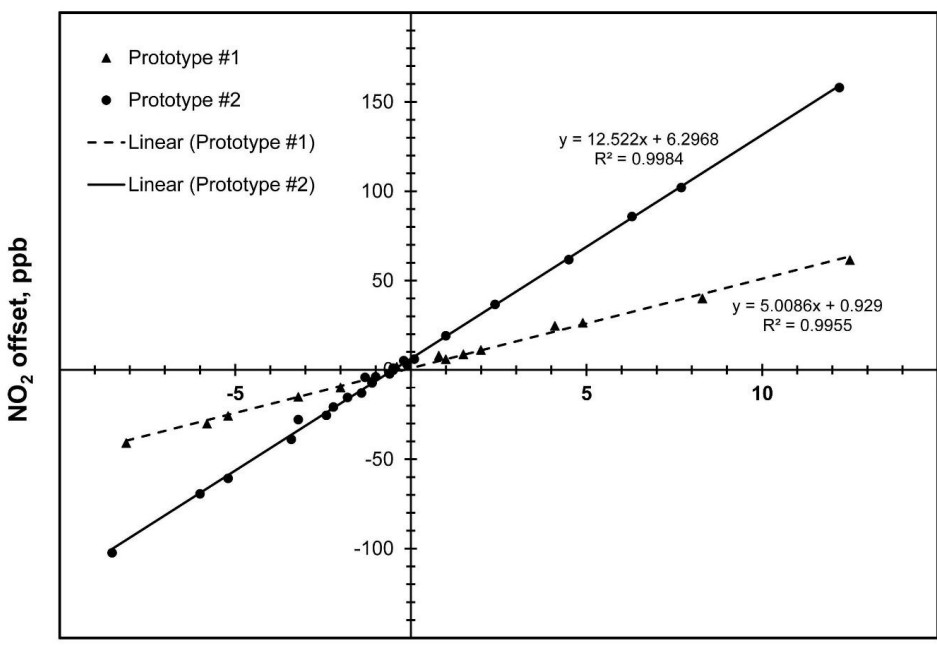

**Figure 4.** Plot of data obtained for two prototypes of the Folded Tubular Photometer showing the analyzer offset in ppb of $NO_2$ as a function of the measured pressure difference ($P_I - P_{I_o}$) between sample bypassing the $NO_x$ scrubber ($I$) and sample passing through the $NO_x$ scrubber ($I_o$).





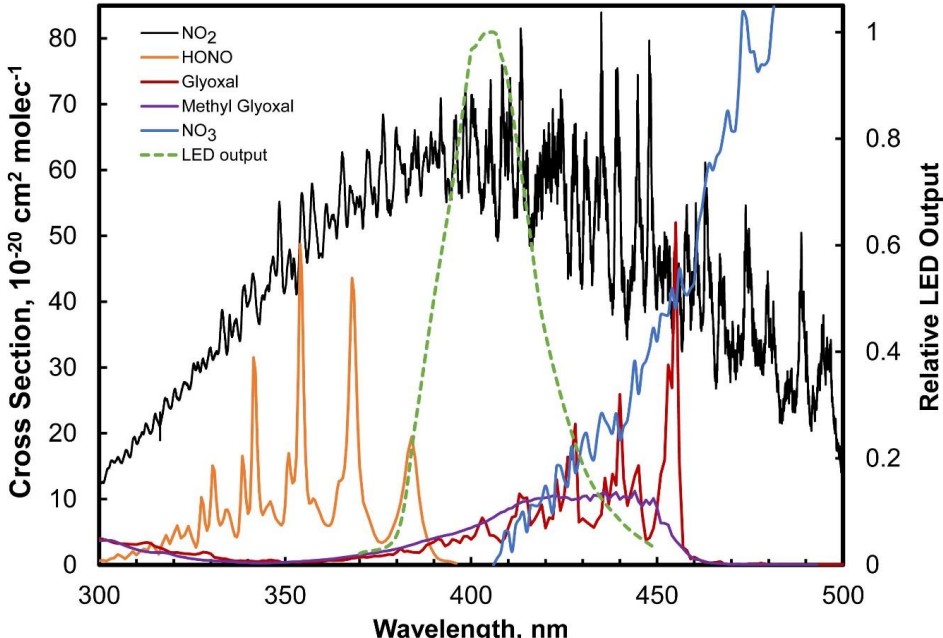

**Figure 5.** Absorption spectra of NO₂ and possible airborne interferences (HONO, NO₃, glyoxal, methyl glyoxal) along with the spectral output of the LED used in the Model 405 nm.





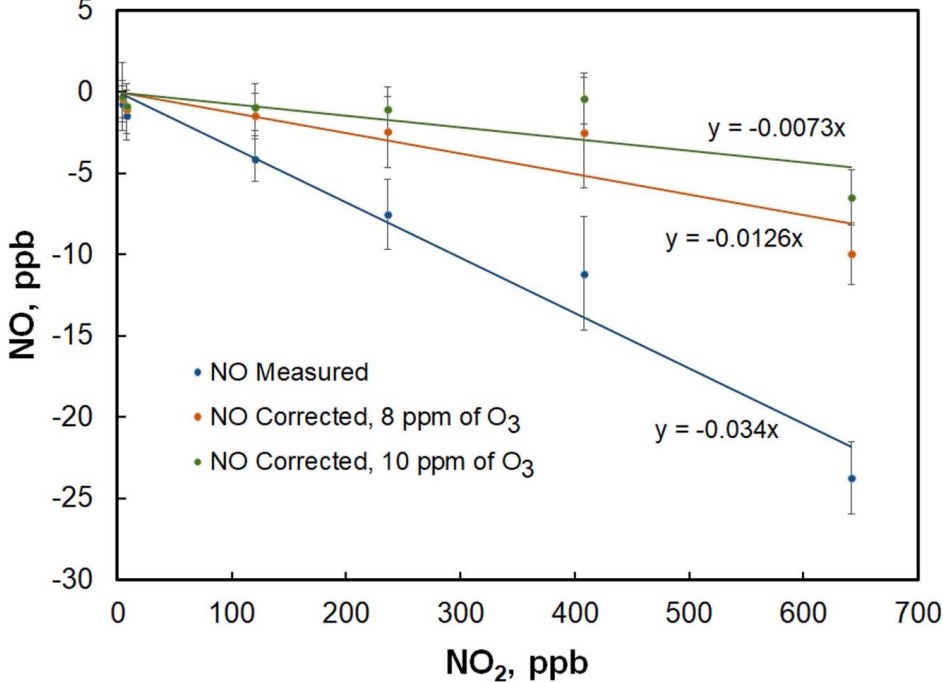

**Figure 6.** Plot of the NO measured in the Folded Tubular Photometer vs. $NO_2$ mixing ratio. The blue line is a linear fit to the data points and yields a slope of -3.4 ppb NO/100 ppb of $NO_2$. Corrected NO concentrations after application of Eq. (6) yield the orange and green points using $O_3$ concentrations of 8 and 10 ppm, respectively. Slopes decrease to -1.3 ppb NO/100 ppb of $NO_2$ and -0.7 ppb NO/100 ppb $NO_2$ for the 8 and 10 ppm cases, respectively.





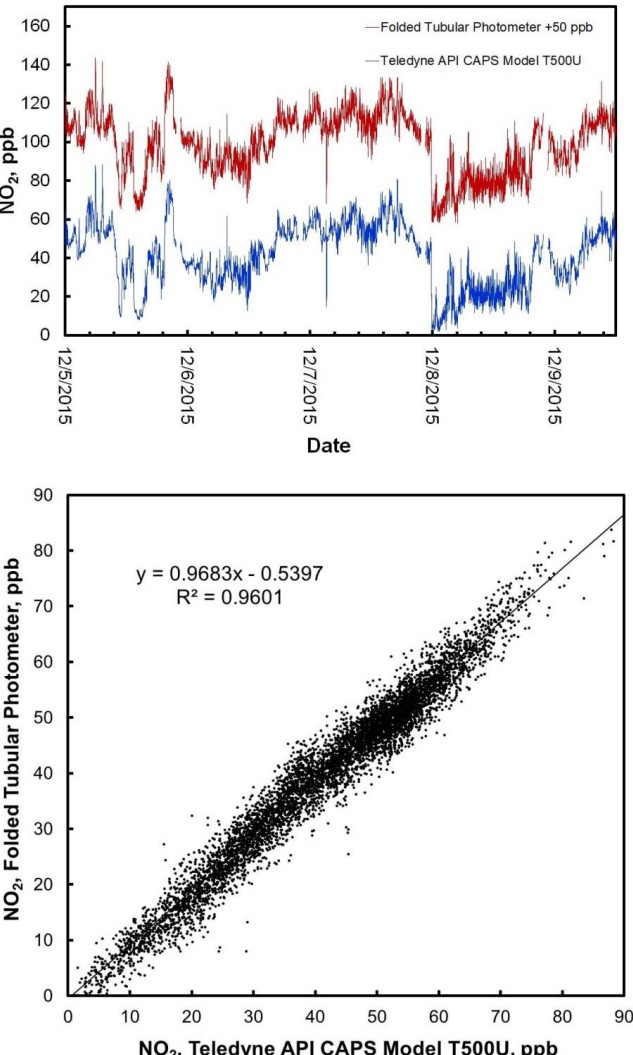

**Figure 7.** Top panel: Time series comparison plot of ambient $NO_2$ concentration measured outdoors
5    at the Colorado Department of Public Health and Environment (CDPHE) Interstate 25/Globeville
roadside site using a Teledyne API Model T500U (lower data line in blue) and a Folded Tubular
Photometer (upper data line in red). Data for the Folded Tubular Photometer are offset for clarity by
adding 50 ppb to the measurements. Bottom panel: The same data, shown as a correlation plot.





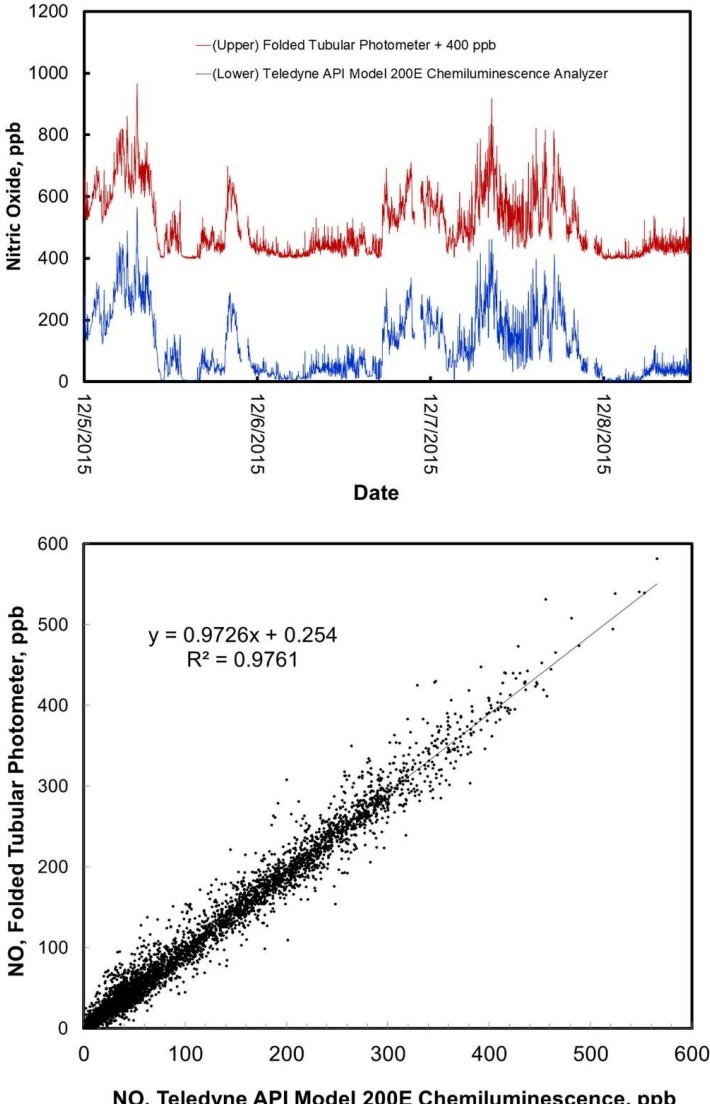

**Figure 8.** Top panel: Time series comparison plot of ambient NO concentration measured outdoors at the CDPHE  Interstate 25/Globeville roadside site using a Teledyne API Model 200E Chemiluminescence Analyzer (lower data line) and a Folded Tubular Photometer (upper data line). Data for the Folded Tubular Photometer are offset for clarity by adding 400 ppb to the measurements. Bottom panel:  The same data, shown as a correlation plot.  Data shown in the bottom panel includes the small correction for $N_2O_5$ formation as described in the text.