# Peer review of "Folded Tubular Photometer for atmospheric measurements of NO2 and NO"

_Atmospheric Measurement Techniques, 2018_

## Referee Comment (RC1) · Anonymous Referee #1 · 26 Feb 2018

The manuscript describes a miniaturised near-UV photometer for quantifying atmospheric NO2, and with conversion by addition of O3, NO by difference. A full description of this commercially available instrument is certainly within the scope of AMT, the level of detail is generally good, as is the presentation quality. I recommend it for publication in AMT after the issues below are addressed.

Major comments:

1) The instrument detects NO2 by single wavelength absorption at 405 nm using an LED light source in a flow cell. The 405 nm region is attractive for NO2 as there are very few other species absorbing strongly in this region. However, this is also the region where the NO2 quantum yield begins to fall to zero from 400 nm upwards. With the relatively broad output of the LED light source the authors present in figure 5 it could

be expected that an LED of sufficiently high output to be reliably detected over the 2.1m path-length, and after passing various mirrors, plus the authors' estimated 90% loss, might have sufficient photolysis power to dissociate a portion of the NO2 intended for analysis. The authors should add the NO2 quantum yield to figure 5 and discuss this uncertainty, or else justify why NO2 photolysis is irrelevant/insignificant. Adding the LED wattage or j value would also help, as would contrasting with current NO2 photolysis systems which use LEDs from 385-395 nm with a similar Gaussian output e.g. Buhr 2007; Pollack et al., 2011; Reed et al., 2016.

2) The unexplained effect of cell pressure on retrieved analyte concentration warrants further investigation. As the effect is different between systems it suggests difference in construction or materials used. Varying water vapour equilibrium concentration should be the same between both prototype instruments assuming variables such as cell temperature and coating hygroscopicity are controlled for. Was the system tested with sample drying? Different leakage rates between systems or changes in cell length under vacuum perhaps are more likely. Never the less some more insight on the phenomena is important.

Minor comments:

1) A low pressure mercury lamp is used for generating ozone; this is presumably by illuminating ambient air though this detail is absent from the text and is assumed by the absence of an O2 bottle.

2) How is the conversion efficiency of NO to NOň22 verified, and is it stable? Low pressure mercury lamps age and reduce in output, chemiluminescence instruments tend to use corona discharge devices for ozone generation due to their stability and high output. Furthermore mercury lamps must be thermos stated to be stable, is this one? Would this negate the advantage of not needing to calibrate the NO2 measurement?

Technical corrections:

P5 L12 – "FTP" – You haven't defined this acronym before using it and do not use it again.

P6 L34 – "In other work, we measured black carbon using an LED with maximum emission near 880 nm." Either show the results, include a reference to that paper, or remove this line as it is left hanging.

P7 L19 - "Voltage Sensitive orifice" The following paragraph describes a pressure controller common to numerous optical absorption systems. Is the name change necessary?

P8 L18 – "...nearly plug-flow..." I'm not sure what nearly plug flow is however looking at Figure 1 suggests that the sample flow is expanded out and made to take a sharp turn at each fold so I find it difficult to believe there is laminar airflow within the system. Remove the reference to plug flow.

P8 L29 – "Scrubber" – provide references or details of this combination of materials being an effective scrubber as this effects the zero bias of the instrument.

P9 L19 – "atm" use bar/mbar consistently as an alternative.

P15 L29 – "...true NO2..." I believe is how the Teledyne T500U and T200U/P instruments are marketed too. Better delete this line as it doesn't add anything.

P16 L9 – insert a space at "...O3resulting..."

References:

Buhr, M.: Solid-State Light Source Photolytic Nitrogen Dioxide Converter, US 7238328 B2, United States, USTPO, available, 3 July 2007.

Pollack, I. B., Lerner, B. M., and Ryerson, T. B.: Evaluation of ultraviolet light-emitting diodes for detection of atmospheric NO2 by photolysis – chemiluminescence, J. Atmos. Chem., 65, 111–125, doi:10.1007/s10874-011-9184-3, 2011.

[Figure]

Reed, C., Evans, M. J., Di Carlo, P., Lee, J. D., and Carpenter, L. J.: Interferences in photolytic NO2 measurements: explanation for an apparent missing oxidant?, Atmos. Chem. Phys., 16, 4707– 4724, doi:10.5194/acp-16-4707-2016, 2016.
* * *

---

## Referee Comment (RC2) · Anonymous Referee #2 · 5 Mar 2018

This manuscript describes an instrument based on near-UV absorption for measuring atmospheric NO2, with NO also measured by conversion to NO2 upon addition of O3. It contains an extensive description of the instrument, potential problems with the measurements and some field data comparison to more 'traditional' measurements of NO2 (CAPs and conversion to NO followed by chemiluminescence detection). The manuscript is certainly within scope of AMT, is well written and gives a very good overview of this new measurement method. I recommend publication in AMT once the following relatively minor issues have been addressed.

General comments:

One of the key parts of the instrument is the 'scrubber' which removes NO and NO2 in order to get the I0 measurement required for the Beer Lambert law calculation of

concentration. This is first mentioned in section 2.1 but with no details of what material is used. The material details are mentioned later in the manuscript but these should be described in the earlier section. There is also no discussion of how efficient the scrubber is or the potential time interval required between scrubber replacement. Any degradation in the efficiency of the scrubber will have a direct detrimental effect on the quality of the NO2 measurements so something should be said about this. Have any tests been carried out on the scrubber in different ambient NO2 levels? The instrument is likely to be most often used in polluted environments and I wonder how well the scrubber works at ambient NO2 levels of 100's of ppb?

NO measurements are made by converting it to NO2 by addition of O3 to the gas flow, with the O3 produce using photolysis of air using a low pressure mercury discharge lamp. The authors state that they get 98.8% conversion within the system. Have they investigated how this conversion might change with lamp age? For how many hours can the instrument be run before a change of O3 lamp is needed. This would seem to be crucial information if the instrument is indeed to be used for long term measurements of NO and NO2.

The authors go into detail about how the accuracy of the instrument is calculated. Could they also make some comment on the detection limit for NO2. This is important information for anyone wanting to use the instrument in more rural or remote environments? By how much could the ultimate sensitivity of the instrument be improved by increasing the pathlength, something that it is stated is possible on P8 L7.

In section 4 the authors describe other techniques (both direct and indirect) for measuring NO2 but Laser Induced Fluorescence (LIF) is not included as a direct technique. Whilst there are no commercially available LIF instrument available it has been used extensively for research with a large amount of literature on the subject so it should at least be briefly mentioned.

On P16 line 22 the authors state that "the cost of both the CAPS and CRDS instru-
ments are significantly higher than the single-pass folded Tubular Photometer described here". I'm not sure this is strictly true, especially with the Teledyne T500U CAPS instrument. Also, CRDS instruments generally has a significantly better detection limit than the instrument described here, which maybe what a user required. I suggest removing reference to the cost of instruments.

The manuscript describes an instrument to measure NO2, however there is a section at the end describing how O3, SO2 and aerosol extinction could be measured by the same technique. Could the authors make some estimates as to how sensitive / accurate such an instrument would be? I also wonder if the text mentioning the other species should really be in the first sentence of the abstract?

The final sentence of the abstract "In contrast to other commercially available direct NO2 measurements, such as cavity-attenuated phase shift spectroscopy (CAPS), the Folded Tubular Photometer provides a means for measuring NO simultaneously in the same apparatus by quantitatively converting NO to NO2 with ozone, which is then detected by direct absorbance.", kind of suggests these other techniques could not measure NO with similar addition of ozone. I don't think this is true - it is just the manufacturers have chosen not to do it. This should be made clear in the abstract and text.

Technical corrections:

P5 L12: The acronym FTP is used here for the only time in the manuscript. The authors should either remove it or use it every time Folded Tubular Photometer is used after the first mention.

P6 L34: Is the a reference to for the black carbon measurement at 880nm?

P8 L5: what material o-rings are used?

- P8 L17: what does 'nearly plug flow' mean?
- P11 L35: Surely the instrument cannot operate in "NO only mode" as the measurement
**would be a sum of NO and NO2? So should this read "NOx or NO2"?**

P16 L9: "O3resulting" space required. Also I think this is the only time O3 is used instead of ozone in the manuscript. The authors should be consistent throughout.

---

## Author Comment (AC1) · 17 Apr 2018

The author responses to both reviewers #1 and #2 are included in the accompanying supplement. This supplement also includes an annotated revised manuscript showing the changes made. We thank both reviewers for their thoughtful comments.

Please also note the supplement to this comment:
https://www.atmos-meas-tech-discuss.net/amt-2018-24/amt-2018-24-AC1-supplement.pdf

---

## Author Response (AR2)

Author Response to the Technical Suggestion from the Editor:

Editor's Comment: *I have now read through the revised manuscript and your responses to the reviewers. Reviewer 2 queried the first sentence of the abstract which refers to the direct*
5 *measurement of SO2, O3 and black carbon. Given that the paper only really focuses on the measurement of NO2 and NO in detail, I think the first sentence of the Abstract needs to be modified to reflect this before publication. I would be happy with something like: '...for making direct measurements of the concentrations of NO2. SO2, O3 and black carbon could also be detected directly by this method.'*

Response: We would like to thank the editor for their suggestion on changing the abstract. As suggested we have changed the first sentence to focus on the measurement of NO2 and then mention that this technique could be extended to measurements of other pollutants. The first sentence now reads: "
[revised manuscript text omitted]